# Learning Human-Robot Collaboration via Heterogeneous-Agent Lyapunov Policy Optimization

Hao Zhang [1 2]   Yaru Niu [2]   Yikai Wang [2]   Ding Zhao [2]   H. Eric Tseng [1]

## Abstract

To improve generalization and resilience in human–robot collaboration (HRC), robots must handle the combinatorial diversity of human behaviors and contexts, motivating multi-agent reinforcement learning (MARL). However, inherent heterogeneity between robots and humans creates a rationality gap (RG), where decentralized policy updates deviate from cooperative joint optimization. The resulting learning problem is a general-sum differentiable game, so independent policy-gradient updates can oscillate or diverge without added structure. We propose heterogeneous-agent Lyapunov policy optimization (HALO), a framework that stabilizes decentralized MARL by enforcing Lyapunov-based contraction in policy-parameter space. Unlike Lyapunov-based safe RL, which targets state/trajectory constraints in constrained Markov decision processes, HALO uses Lyapunov certification to stabilize decentralized policy learning. HALO rectifies decentralized gradients via optimal quadratic projections, ensuring monotonic contraction of RG and enabling effective exploration of open-ended interaction spaces. Extensive simulations and real-world humanoid-robot experiments show that this certified stability improves generalization and robustness in collaborative corner cases.

## 1. Introduction

Human-robot collaboration (HRC) is a central challenge for embodied intelligence in human environments, requiring robots to achieve task-level coordination with diverse and adaptive human, and potentially robotic, partners. (Vinyals et al., 2019). Traditional HRC is framed as a single-agent task where the human is treated as a static or perturbed environmental component (Foerster et al., 2018). Such robot–script or log–replay paradigm relies on simulators with predefined human inputs, failing to capture the stochastic richness in human behaviors (Hadfield-Menell et al., 2016). Consequently, robots often overfit to specific interaction traces (Carroll et al., 2019), leading to performance collapse when encountering out-of-distribution (OOD) behaviors (Samvelyan et al., 2019; Strouse et al., 2021).

To transcend these limits, this work adopts heterogeneous multi-agent reinforcement learning (MARL) for human–robot synergy (Oroojlooy & Hajinezhad, 2023). We argue - and later demonstrated empirically in our experiment - that replacing static scripts with learning-capable humanoid proxies as a computational imperative (Haight et al., 2025), enabling robots to navigate infinite interaction manifolds (Lowe et al., 2017). MARL allows adaptive strategies to emerge that are intractable via manual scripting (Li et al., 2025). This ensures that complex edge cases are captured (Hu et al., 2020), providing a foundation for generalization. However, heterogeneous learning introduces a critical structural pathology known as the rationality gap (RG) (Kim et al., 2021). In MARL, agents share a team-level objective, but heterogeneity forces each agent to update from its own individual perspective (Rashid et al., 2020; Kang et al., 2023). While many prior MARL methods rely on parameter-sharing that collapse the joint parameter space into a shared representation (Wen et al., 2022), such sharing is infeasible in heterogeneous HRC settings. This mismatch further widens the RG, as individual updates diverge from team-optimal directions (Son et al., 2019).

Beyond this structural misalignment, decentralized learning also suffers from inherent dynamical instabilities. MARL updates evolve under a non-conservative vector field with a non-symmetric Jacobian, giving rise to rotational dynamics and limit cycles (Zhao et al., 2023; Balduzzi et al., 2018; Letcher et al., 2018). Prior work in differentiable games has proposed several methods to damp or compensate for these rotational forces, including symplectic gradient adjustment, which subtracts the antisymmetric component of the Jaco-

[1]ETAIC Lab, Department of Electrical Engineering, University of Texas at Arlington, USA [2]Safe AI Lab, Department of Mechanical Engineering, Carnegie Mellon University, USA. Correspondence to: H. Eric Tseng <hongtei.tseng@uta.edu>, Ding Zhao <dingzhao@andrew.cmu.edu>, Hao Zhang <haoz4@andrew.cmu.edu>.

*Proceedings of the $43^{rd}$ International Conference on Machine Learning*, Seoul, South Korea. PMLR 306, 2026. Copyright 2026 by the author(s).

bian to reduce cycling (Balduzzi et al., 2018). Consensus optimization and its variants that regularize gradients toward more potential-like behavior (Mescheder et al., 2017), extragradient and optimistic methods that stabilize saddle-point dynamics (Gidel et al., 2018; Daskalakis et al., 2017), and opponent-shaping approaches that estimate how an agent's update will influence others (Foerster et al., 2017). However, these techniques typically assume low-dimensional differentiable games, centralized Jacobian access, or fully modeled opponents, and thus remain difficult to apply in heterogeneous, partially observed, embodied HRC settings. As a result, heterogeneous agents often still "chase" one another's evolving strategies, producing unstable oscillations that prevent convergence to cooperative optima (Mazumdar et al., 2020; Fiez et al., 2020) leaving exploration largely tethered to a non-convergent regime (Yang et al., 2024).

Consequently, existing HRC architectures lack a stability kernel capable of neutralizing these non-conservative forces (Chow et al., 2018). To the best of the authors' knowledge, the integration of MARL-based interaction paradigm with a learning-stability kernel remains an open challenge in the context of HRC (Gu et al., 2023). Therefore, we introduce heterogeneous-agent Lyapunov policy optimization (HALO), which establishes a formal stability certificate in the policy-parameter space by quantifying coordination disagreement as a Lyapunov potential. By employing an optimal quadratic projection to rectify optimization dynamics, HALO ensures the monotonic contraction of the RG.

Our contributions are summarized as follows: 1) we propose a learning kernel HALO that enforces stable policy-parameter updates via an optimal quadratic projection, yielding a formal stability certificate in parameter space; 2) We establish theoretical guarantees, proving monotonic contraction of the rationality gap under HALO using nonlinear stability analysis (Khalil & Grizzle, 2002); 3) we demonstrate HALO across diverse HRC tasks and formalize why autonomous exploration with HALO is necessary to avoid the OOD brittleness of scripted HRC.

## 2. Related Work

**Learning paradigms for HRC.** Conventional HRC treats humans as reactive environment components via predefined scripts (Jaderberg et al., 2019), limiting coordination to finite interaction patterns (Foerster et al., 2018). Such single-agent formulations or imitation learning fail to generalize to non-stationary human behaviors (Vinyals et al., 2019; Raileanu et al., 2018). Therefore, the transition to co-adaptation is imperative for handling latent human intentions (Sarkadi et al., 2018). This work circumvents this by replacing scripts with learning-capable humanoid agents, and using MARL to force the robot to internalize a broader distribution of coordination patterns (Strouse et al., 2021).

**Stability in MARL.** MARL instability stems from differentiable game dynamics, where non-symmetric Jacobians and solenoidal vector fields induce rotational behaviors that obstruct convergence (Balduzzi et al., 2018; Zhao et al., 2023; Kim et al., 2021). Centralized training with decentralized execution (CTDE) methods address this via value factorization (Rashid et al., 2020; Son et al., 2019; Wang et al., 2020) or trust-region heuristics (Gu et al., 2021), yet they regularize update magnitudes rather than geometric directions. In contrast, our HALO algorithm analyzes the Lyapunov descent condition to neutralize the cyclic divergence in heterogeneous gradients (Fiez et al., 2020).

**Lyapunov methods in RL.** In safe RL, Lyapunov functions are used as certificates to enforce constraint-satisfaction conditions during learning (Chow et al., 2018), sometimes augmented by additional safety tools, including barrier function (Sikchi et al., 2021). More broadly, Lyapunov-based tools have been used to ensure stability of learned dynamics models (Kolter & Manek, 2019) and to infer stability certificates directly from data, extending a long tradition in nonlinear systems analysis (Boffi et al., 2021). Despite these advancements, there is little exploration of using Lyapunov functions to directly certify the stability of policy-parameter learning dynamics in MARL (Leonardos et al., 2021). This work moves in this direction by applying Lyapunov principles on policy-parameter space, inducing a contracting potential even under non-stationary updates.

**Gradient alignment and geometry.** Geometric heuristics like PCGrad (Yu et al., 2020) mitigate conflicts by projecting gradients with negative similarity, yet they lack global invariants over the learning trajectory. More robust geometric approaches involve Riemann-Finsler metrics (Yang & Nachum, 2021) or mirror descent on the simplex (Shani et al., 2020). Frameworks like heterogeneous mirror learning provide unified convergence guarantees for multi-objective settings (Zhong et al., 2024). Our HALO extends geometric intuitions toward a Lyapunov-based perspective on learning dynamics, using stability principles to formalize contraction properties that promote coordination. The full algorithmic details appear in the following section.

## 3. Preliminaries

### 3.1. Decentralized POMDPs

HRC tasks use a decentralized partially observable Markov decision process (POMDP) $\mathcal{M} = \langle \mathcal{S}, \mathcal{A}, P, R, \gamma, N, \mathcal{O}, Z \rangle$ (Foerster et al., 2018). Unlike parameter sharing, heterogeneous agents rely on independent policies $\pi_{\theta_i}(a_{i,t}|o_{i,t})$ given local observations $o_{i,t} = Z(s_t, i)$. The parameter vector is defined as $\theta = [\theta_1^\top, \ldots, \theta_N^\top]^\top \in \mathbb{R}^D$. Agents share a global reward $r_t = R(s_t, \mathbf{a}_t)$ and the objective is to maximize the return $J(\theta) = \mathbb{E}_{\mathbf{a}_t \sim \pi_\theta, s_t \sim P}[\sum_{t=0}^\infty \gamma^t R(s_t, \mathbf{a}_t)]$.

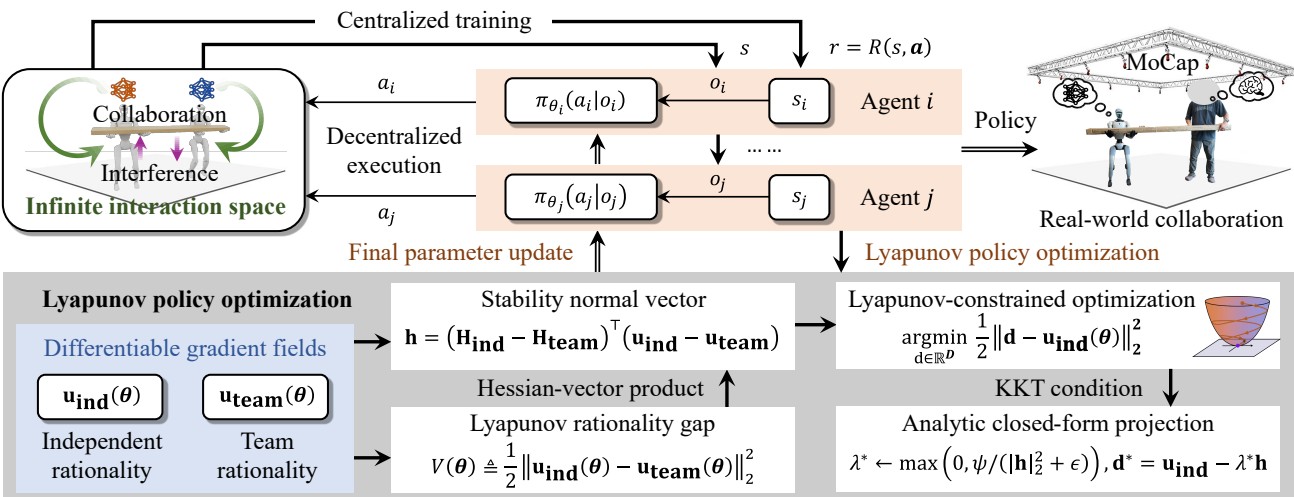

*Figure 1.* The HALO framework architecture combining the transition from standard decentralized learning to Lyapunov policy optimization for real-world HRC. Key components include the computation of the rationality gap $V(\theta)$ and the stability normal vector $h$ to derive the final analytic closed-form projection $d^*$.

### 3.2. Decoupled CTDE and the stationarity assumption

Under the CTDE paradigm (Yu et al., 2020), each agent independently updates its parameters for heterogeneous embodiments, where $\hat{A}_{tot}$ is a centralized advantage estimator:
$\nabla_{\theta_i} J_i(\theta_i) = \mathbb{E}\left[\nabla_{\theta_i} \log \pi_{\theta_i}(a_i|o_i)\hat{A}_{tot}(s, \mathbf{a})\right]$. The concatenation of these updates forms an independent rationality field $[\nabla_{\theta_1} J_1^\top, \ldots, \nabla_{\theta_N} J_N^\top]^\top$, computed as if the partner's policies $\pi_{\theta_{-i}}$ were part of a fixed environment component. This implicitly assumes partner stationarity.

### 3.3. Learning dynamics and rationality gap

A fundamental pathology in decoupled architectures is that $\mathbf{u}_{\text{ind}}(\theta)$ constitutes a non-conservative vector field (Balduzzi et al., 2018). Because agent parameters are independently updated, the joint Jacobian is non-symmetric, $\nabla_{\theta_j}\nabla_{\theta_i} J_i \neq \nabla_{\theta_i}\nabla_{\theta_j} J_j$, inducing rotational components that lead to limit cycles and divergent trajectories (Zhao et al., 2023). The structural mismatch in HRC creates a rationality gap between the decentralized update directions and the true team-level ascent direction $\nabla J(\theta)$.

## 4. Methodology: The HALO Framework

We design a stability-aware control law that rectifies decentralized gradients to satisfy a convergence certificate, as illustrated in Fig. 1.

### 4.1. Vector field misalignment and Lyapunov stability

Let $\theta = [\theta_1^\top, \ldots, \theta_N^\top]^\top \in \mathbb{R}^D$ represent the joint parameter vector of $N$ heterogeneous agents. In the CTDE paradigm with decoupled architectures (Zhao et al., 2023), the learning

dynamics are governed by the interaction between local agent intentions and the global team objective. We formalize this interaction via two competing vector fields:

1. The independent rationality field ($\mathbf{u}_{\text{ind}}$): This field is formed by the concatenation of individual actor gradients. For each agent $i$, the update is driven by a local surrogate $J_i(\theta_i) = \mathbb{E}_{a_i \sim \pi_{\theta_i}}[Q_{\text{tot}}(s, \mathbf{a})]$, which assumes other agents' policies are momentarily stationary:

$$\mathbf{u}_{\text{ind}}(\theta) \triangleq [\nabla_{\theta_1} J_1^\top, \ldots, \nabla_{\theta_N} J_N^\top]^\top \in \mathbb{R}^D. \quad (1)$$

2. The team rationality field ($\mathbf{u}_{\text{team}}$): This represents the true ascent direction of the global team reward function $J(\theta) = \mathbb{E}_{\mathbf{a} \sim \pi_\theta}[\sum_t \gamma^t r_t]$. Under the chain rule in the joint parameter space, it defines the team rationality field:

$$\mathbf{u}_{\text{team}}(\theta) \triangleq \nabla_\theta J(\theta) = \left[\frac{\partial J}{\partial \theta_1}^\top, \ldots, \frac{\partial J}{\partial \theta_N}^\top\right]^\top \in \mathbb{R}^D. \quad (2)$$

In this formulation, we define the rationality gap, the Variational mismatch between decentralized best-response dynamics and centralized cooperative dynamics, via a Lyapunov candidate function as the discrepancy:

$$V(\theta) \triangleq \frac{1}{2}\|\mathbf{u}_{\text{ind}}(\theta) - \mathbf{u}_{\text{team}}(\theta)\|_2^2. \quad (3)$$

Structural pathology: In heterogeneous MARL, $\mathbf{u}_{\text{ind}}$ is generally non-conservative (Balduzzi et al., 2018). With decoupled parameters, the Jacobian $\mathbf{H}_{\text{ind}} \triangleq \nabla_\theta \mathbf{u}_{\text{ind}}$ is non-symmetric, as cross-terms $\nabla_{\theta_j}\nabla_{\theta_i} J_i$ and $\nabla_{\theta_i}\nabla_{\theta_j} J_j$ differ. By Helmholtz decomposition, $\mathbf{u}_{\text{ind}} = \nabla\Phi + \Psi$, where the solenoidal component $\Psi$ drives limit cycles and oscillations

(Zhao et al., 2023). $V(\theta)$ monitors this dissonance and our control objective is to design an update $\mathbf{d}$ realizing

$$\langle \nabla_\theta V, \mathbf{d} \rangle \leq -\sigma V(\theta), \quad \sigma > 0, \qquad (4)$$

enforcing a Lyapunov dissipation constraint for asymptotic contraction of the rationality gap and stabilizing decentralized learning.

### 4.2. Structural stability and analytic projection

Standard decentralized updates $\theta_{k+1} = \theta_k + \eta \mathbf{u}_{\text{ind}}$ prioritize local greedy progress but frequently increase the rationality gap $V(\theta)$ (Dai et al., 2025). To ensure structural stability, we seek an optimal update direction $\mathbf{d}^*$ that strictly satisfies a Lyapunov stability certificate (Yang et al., 2020). This is formulated as a constrained quadratic program:

$$\min_{\mathbf{d} \in \mathbb{R}^D} \quad \frac{1}{2} \|\mathbf{d} - \mathbf{u}_{\text{ind}}(\theta_k)\|_2^2 \qquad (5)$$
$$\text{s.t.} \quad \langle \nabla_\theta V(\theta_k), \mathbf{d} \rangle \leq -\sigma V(\theta_k).$$

Eq. (5) performs a minimum-norm projection of $\mathbf{u}_{\text{ind}}$ onto the stability half-space $\mathcal{H}_{\text{stable}} = \{\mathbf{d} \in \mathbb{R}^D \mid \nabla V^\top \mathbf{d} \leq -\sigma V\}$. This functions as a structural stability certificate based on decentralized gradients. The Karush-Kuhn-Tucker (KKT) conditions permit an exact analytic solution (Clempner, 2016). Let $\mathbf{h} \triangleq \nabla_\theta V(\theta_k)$ denote the gradient of the disagreement and we define the Lagrangian as:

$$\mathcal{L}(\mathbf{d}, \lambda) = \frac{1}{2} \|\mathbf{d} - \mathbf{u}_{\text{ind}}\|_2^2 + \lambda \left( \mathbf{h}^\top \mathbf{d} + \sigma V \right), \qquad (6)$$

where $\lambda$ is the dual variable. For optimal update $\mathbf{d}^*$, the stationarity condition $\nabla_\mathbf{d} \mathcal{L} = 0$, combined with the primal feasibility $\mathbf{h}^\top \mathbf{d}^* + \sigma V \leq 0$ and complementary slackness $\lambda(\mathbf{h}^\top \mathbf{d}^* + \sigma V) = 0$, reveals the structure $\mathbf{d}^* = \mathbf{u}_{\text{ind}} - \lambda \mathbf{h}$. To determine the optimal multiplier $\lambda^*$, we substitute the stationarity condition into the slackness equation:

$$\lambda \left( \mathbf{h}^\top (\mathbf{u}_{\text{ind}} - \lambda \mathbf{h}) + \sigma V \right) = 0$$
$$\implies \lambda(\mathbf{h}^\top \mathbf{u}_{\text{ind}} - \lambda \|\mathbf{h}\|_2^2 + \sigma V) = 0. \qquad (7)$$

Solving Eq. (7) identifies two operational regimes: an inactive regime where $\mathbf{h}^\top \mathbf{u}_{\text{ind}} + \sigma V \leq 0$ (yielding $\lambda^* = 0$). Unifying these cases via the rectifier function yields the HALO projection operator, where $\epsilon$ is a damping constant:

$$\mathbf{d}^* = \mathbf{u}_{\text{ind}} - \max\left( 0, \frac{\langle \mathbf{h}, \mathbf{u}_{\text{ind}} \rangle + \sigma V}{\|\mathbf{h}\|_2^2 + \epsilon} \right) \mathbf{h} \qquad (8)$$

### 4.3. Scalability via Hessian-vector product

A primary concern regarding Lyapunov-based optimization is the perceived second-order complexity. The vector $\mathbf{h} =$

---

**Algorithm 1** HALO practical implementation

---

**Require:** Initial joint policy $\theta_0$, critic $Q_\phi$, hyper-parameters $\eta, \sigma, \epsilon$
1: **for** iteration $k = 0, 1, \ldots$ **do**
2:     Sample mini-batch $\mathcal{D} \sim \pi_{\theta_k}$
3:     {*Step 1: Compute differentiable gradient fields*}
4:     $\mathbf{u}_{\text{ind}} \leftarrow \nabla_\theta \mathcal{L}_{\text{ind}}|_{\theta_k}$ with `create_graph=True`
5:     $\mathbf{u}_{\text{team}} \leftarrow \nabla_\theta \mathcal{L}_{\text{team}}|_{\theta_k}$ with `create_graph=True`
6:     {*Step 2: Obtain stability normal $\mathbf{h}$ via HVP*}
7:     $V \leftarrow \frac{1}{2} \|\mathbf{u}_{\text{ind}} - \mathbf{u}_{\text{team}}\|_2^2$
8:     $\mathbf{h} \leftarrow \nabla_\theta V|_{\theta_k}$        ▷ *Double back-prop pass*
9:     {*Step 3: Stability-constrained projection*}
10:    $\psi \leftarrow \langle \mathbf{h}, \text{detach}(\mathbf{u}_{\text{ind}}) \rangle + \sigma \cdot \text{detach}(V)$
11:    $\lambda^* \leftarrow \max\left( 0, \psi/(\|\mathbf{h}\|_2^2 + \epsilon) \right)$
12:    {*Step 4: Update parameters and critic*}
13:    $\mathbf{d}^* \leftarrow \text{detach}(\mathbf{u}_{\text{ind}}) - \lambda^* \mathbf{h}$
14:    $\theta_{k+1} \leftarrow \theta_k + \eta \mathbf{d}^*$
15:    Update $\phi$ by minimizing $\mathcal{L}_{\text{MSE}}(Q_{\text{tot}}, y)$
16: **end for**

---

$\nabla_\theta V$ requires differentiating through the gradient fields, involving a Jacobian-vector product:

$$\mathbf{h} = \nabla_\theta \left( \frac{1}{2} \|\mathbf{u}_{\text{ind}} - \mathbf{u}_{\text{team}}\|_2^2 \right)$$
$$= (\mathbf{H}_{\text{ind}} - \mathbf{H}_{\text{team}})^\top (\mathbf{u}_{\text{ind}} - \mathbf{u}_{\text{team}}), \qquad (9)$$

where $\mathbf{H}$ denotes the Jacobian of the respective vector fields. While explicit $\mathcal{O}(D^2)$ Hessian construction is intractable, HALO leverages double back-propagation to compute Eq. (9) as a Hessian-vector product (HVP). This procedure, detailed in **Algorithm 1**, by retaining the computational graph, the backward pass on $V$ yields the required product without ever materializing the full Hessian matrix. The detailed step-by-step derivation is provided in **Appendix A.1**.

## 5. Theoretical Analysis

**Assumption 5.1** (Regularity and smoothness). The team objective $J(\theta)$ is $C^2$-continuous and the Lyapunov potential $V(\theta)$ is $L$-smooth on the parameter manifold $\Theta$. That is, $\|\nabla_\theta V(\theta_1) - \nabla_\theta V(\theta_2)\|_2 \leq L\|\theta_1 - \theta_2\|_2$ for all $\theta_1, \theta_2 \in \Theta$.

### 5.1. Monotonic descent of the rationality gap

HALO transforms a potentially oscillatory decentralized learning process into a dissipative dynamical system.

**Theorem 5.2** (Monotonicity of potential decay). *Under Assumption 5.1, let $\{\theta_k\}_{k=0}^\infty$ be the sequence of parameters generated by the HALO update law. If the learning rate $\eta$ satisfies the stability bound $\eta \leq 2\sigma V(\theta_k)/(L\|\mathbf{d}_k^*\|_2^2)$, then*

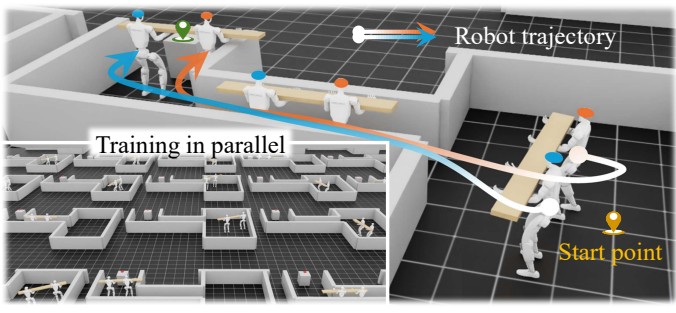

*(a)* Simulation infrastructure and task snapshots.

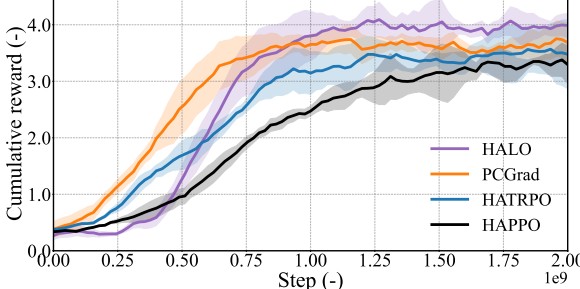

*(b)* Learning dynamics (cumulative reward).

*Figure 2.* Simulation benchmark and learning dynamics: (a) massively parallelized training infrastructure in Isaac Lab, where the arrows indicate the emergent synergy collaboration; (b) performance comparison across nine scenarios, where HALO demonstrates significantly faster convergence, reaching its performance plateau at approximately 1.3B steps.

*the rationality gap $V(\theta)$ is monotonically non-increasing:*

$$V(\theta_{k+1}) - V(\theta_k) \leq -\eta\sigma V(\theta_k) + \frac{L\eta^2}{2}\|\mathbf{d}_k^*\|_2^2. \quad (10)$$

*Summary.* The proof utilizes the descent Lemma for $L$-smooth functions. By solving the KKT conditions for the stability-constrained projection, we show that the update direction $\mathbf{d}_k^*$ always maintains a dissipative inner product with the stability gradient, $\langle \nabla V, \mathbf{d}^* \rangle \leq -\sigma V$. The detailed algebraic derivation is provided in **Appendix A.2**.  □

### 5.2. Asymptotic convergence to equilibrium

Beyond local stability, we establish that HALO drives the multi-agent system toward a state of rationality agreement.

**Theorem 5.3** (Convergence to the synergy manifold). *Suppose $V(\theta)$ is bounded below by 0 and the learning rate $\{\eta_k\}$ satisfies the Robbins-Monro conditions ($\sum \eta_k = \infty, \sum \eta_k^2 < \infty$). Then, the sequence of disagreement energies $\{V(\theta_k)\}_{k=0}^{\infty}$ converges to zero. Consequently:*

$$\lim_{k\to\infty} \|\mathbf{u}_{ind}(\theta_k) - \mathbf{u}_{team}(\theta_k)\|_2 = 0. \quad (11)$$

*Summary.* We construct a summable sequence of the potential energies and apply the monotone convergence theorem. The divergence of $\sum \eta_k$ ensures that the Rationality Gap must vanish asymptotically. The convergence $V \to 0$ implies that the limit points of HALO are stationary points where decentralized preferences $\nabla_{\theta_i} J_i$ are aligned with global team ascent directions. The complete measure-theoretic treatment is provided in **Appendix A.3**.  □

## 6. Experiments and Results

### 6.1. Experimental setup

**Embodied task suite and test setup.** We study three continuous-space coordination tasks: (1) **Orientation-sensitive pushing (OSP)**: pushing an object through a

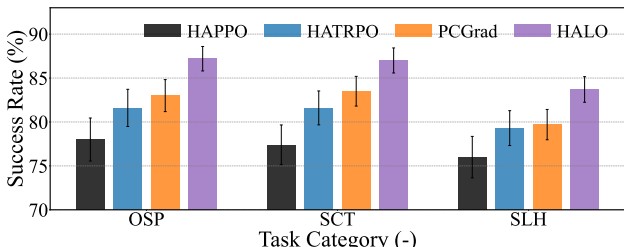

*Figure 3.* Comparison of HALO and baseline MARL algorithms across the nine scenarios in OSP, SCT and SLH tasks.

directional opening requiring precise yaw alignment. (2) **Spatially-confined transport (SCT)**: transport through narrow passages requiring synchronized velocity and tight spatial coordination. (3) **Super-long object handling (SLH)**: transporting a long board via coordinated pivoting and shuffling maneuvers. The training is performed in the Isaac Lab (Mittal et al., 2023), and the physical experiments are conducted on a Unitree G1 robot cooperating with a human partner with reliance on motion-capture (MoCap) system. The detailed test settings are provided in **Appendix B**.

**Baselines and metrics.** We evaluate HALO against state-of-the-art heterogeneous MARL methods: (1) **HAPPO** and **HATRPO**, representing the sequential trust-region paradigm; (2) **PCGrad**, a baseline integrates the HAPPO architecture with gradient surgery. Performance is quantified via success rate (SR), gradient alignment $\cos(\phi)$ (Align), the rationality gap $V(\theta)$ (Gap) and gradient conflict rate (GCR) as the primary stability indicator.

### 6.2. Performance benchmark in simulation

We demonstrate the performance superiority of HALO across physical coupling tasks including OSP, SCT and SLH, shwon in Fig. 2a and the cumulative reward is visualized in Fig. 2b. Besides, Fig. 3 further illustrates the scenario-specific success rates. As synthesized in Table 1, in

*Table 1.* Comprehensive performance matrix and global optimization analysis of heterogeneous coordination: the upper part evaluates the scenario-specific success rate across nine representative coordination challenges in OSP, SCT and SLH tasks, reported as mean $\pm$ std. The lower part provides a synchronized mechanism analysis at the 2B-step steady state, correlating overall task proficiency with fundamental optimization metrics including Overall success rate, convergence, final return, gradient alignment ($\cos \phi$), rationality gap ($V$) and gradient conflict rate. Bold and underlined indicate first and second best, respectively.

| Task category | Scenario name | HAPPO | HATRPO | PCGrad | HALO (Ours) | Improvement |
|---|---|---|---|---|---|---|
| **OSP** | Alignment | $83.6 \pm 5.3$ | $\underline{87.9 \pm 4.5}$ | $87.7 \pm 3.8$ | $\mathbf{92.8 \pm 3.0}$ | +5.6% |
| | Turnaround | $77.6 \pm 6.3$ | $81.5 \pm 5.5$ | $\underline{83.6 \pm 4.8}$ | $\mathbf{86.7 \pm 3.5}$ | +3.7% |
| | Corner entry | $72.7 \pm 7.0$ | $75.3 \pm 6.0$ | $\underline{77.8 \pm 5.3}$ | $\mathbf{82.2 \pm 4.0}$ | +5.7% |
| | OSP average | $78.0 \pm 6.3$ | $81.6 \pm 5.3$ | $\underline{83.0 \pm 4.5}$ | $\mathbf{87.2 \pm 3.5}$ | **+5.1%** |
| **SCT** | Narrow gate | $80.1 \pm 4.8$ | $83.4 \pm 4.3$ | $\underline{88.6 \pm 3.5}$ | $\mathbf{91.1 \pm 2.8}$ | +2.8% |
| | S-shaped path | $76.3 \pm 5.8$ | $\underline{82.1 \pm 5.0}$ | $80.6 \pm 4.5$ | $\mathbf{84.9 \pm 3.8}$ | +3.4% |
| | U-shaped path | $75.7 \pm 6.5$ | $79.2 \pm 5.3$ | $\underline{81.3 \pm 4.8}$ | $\mathbf{85.1 \pm 4.3}$ | +4.7% |
| | SCT average | $77.4 \pm 5.8$ | $81.6 \pm 4.8$ | $\underline{83.5 \pm 4.3}$ | $\mathbf{87.0 \pm 3.5}$ | **+4.2%** |
| **SLH** | Facing mode | $79.3 \pm 5.0$ | $\underline{84.4 \pm 1.6}$ | $83.2 \pm 3.5$ | $\mathbf{88.2 \pm 2.8}$ | +4.5% |
| | Lateral shuffle | $73.7 \pm 6.8$ | $75.9 \pm 5.8$ | $\underline{77.3 \pm 5.0}$ | $\mathbf{80.7 \pm 4.5}$ | +4.4% |
| | Pivoting | $74.9 \pm 6.0$ | $77.5 \pm 5.3$ | $\underline{78.6 \pm 4.5}$ | $\mathbf{82.3 \pm 3.8}$ | +4.7% |
| | SLH average | $76.0 \pm 6.0$ | $79.3 \pm 5.0$ | $\underline{79.7 \pm 4.3}$ | $\mathbf{83.7 \pm 3.8}$ | **+5.0%** |

| **Learning stability and mechanism analysis (steady-state at 2B steps)** | | | | | |
|---|---|---|---|---|---|
| Algorithm | Overall SR $\uparrow$ | Conv. step $\downarrow$ | Final return $\uparrow$ | Align $\cos \phi \uparrow$ | Gap $V \downarrow$ | GCR $\downarrow$ |
| **HAPPO** | 77.1% | 1.8B | 3.44 | 0.67 | 4.89 | 72.5% |
| **HATRPO** | 80.8% | 1.3B | 3.60 | 0.68 | 1.53 | 54.2% |
| **PCGrad** | $\underline{82.1\%}$ | $\mathbf{1.2B}$ | $\underline{3.76}$ | $\underline{0.84}$ | $\underline{0.20}$ | $\underline{25.0\%}$ |
| **HALO** | $\mathbf{86.0\%}$ | 1.3B | $\mathbf{4.02}$ | $\mathbf{0.91}$ | $\mathbf{0.09}$ | $\mathbf{4.2\%}$ |

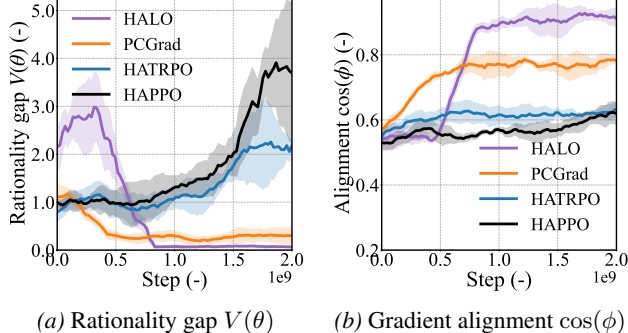

*(a)* Rationality gap $V(\theta)$  *(b)* Gradient alignment $\cos(\phi)$

*Figure 4.* Optimization dynamics analysis: (a) monotonic dissipation of $V(\theta)$ under the Lyapunov stability certificate; (b) rapid convergence of gradient alignment. HALO eliminates solenoidal components to stabilize the joint parameter manifold.

the OSP task, HALO achieves an average SR of 87.2%, outperforming HATRPO (81.6%) and HAPPO (78.0%). These results are consistent with our structural pathology analysis, with HALO achieving a rationality gap of 0.09 and the highest alignment score of 0.91. The increased computation time of HVP is diluted by environment sampling and inference in high-throughput settings, with negligible memory overhead ($< 21.1\%$) due to autograd-based HVP without explicit Hessian materialization, and HALO achieves a 13.1% reduction

in total wall-clock training time compared to HAPPO.

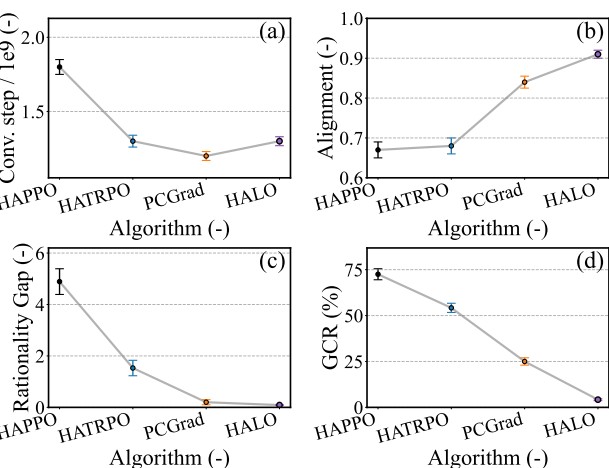

*Figure 5.* Scalability and algorithm metrics analysis: (a) convergence steps required to reach performance plateau; (b) steady-state rationality gap $V$; (c) final gradient alignment $\cos \phi$. (d) gradient conflict rate across algorithms.

### 6.3. Mechanism analysis of HALO

**Geometric rectification of the vector field.** As shown in Fig. 4(a), Fig. 5 and Table 1, HALO ensures a descent of

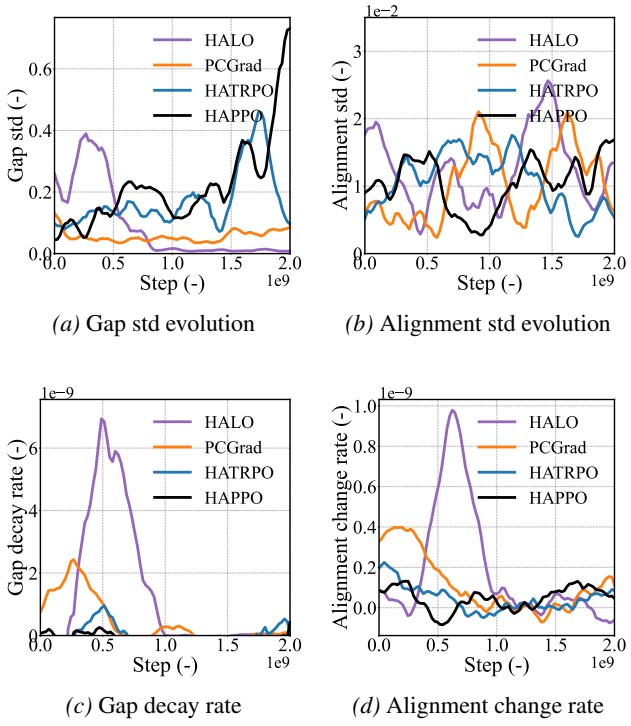

*(a)* Gap std evolution     *(b)* Alignment std evolution

*(c)* Gap decay rate     *(d)* Alignment change rate

*Figure 6.* Detailed mechanism evolution: (a) standard deviation of rationality gap; (b) standard deviation of alignment; (c) temporal decay rate of the gap; (d) instantaneous change rate of alignment.

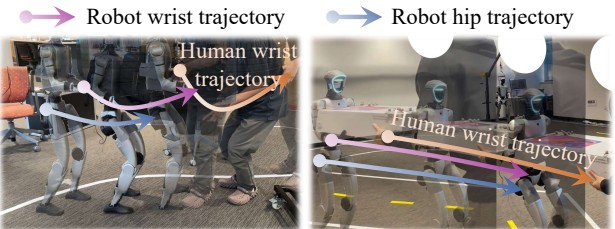

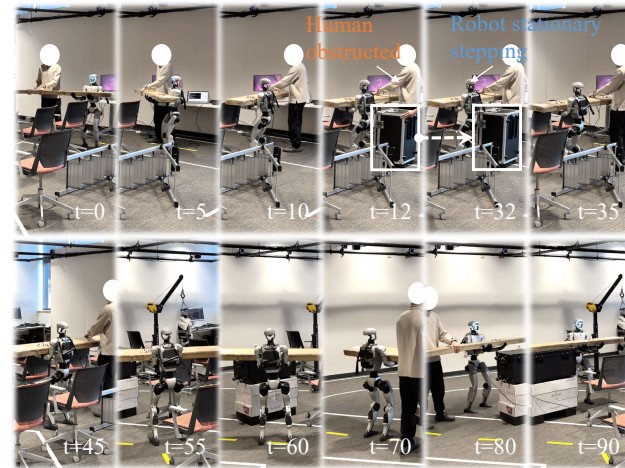

*(a)* Vertical synchronization: adaptive squatting

*(b)* Movement synchronization: obstruction resilience

*Figure 7.* Micro-view analysis of coordination resilience: (a) reactive height modulation during unscripted partner motion; (b) stability maintenance during 20s obstructions through stationary stepping and velocity re-synchronization.

$V(\theta)$, reaching a steady state of 0.09, while HAPPO shows a gap of 4.89. This is further evidenced by the temporal decay rate of the gap achieved by HALO (Fig. 6).

**Stabilization via alignment.** Table 1 demonstrates that by projecting $\mathbf{u}_{\text{ind}}$ onto the stability half-space $\mathcal{H}_{\text{stable}}$, HALO achieves a global alignment of 0.91 and reduces the GCR to 4.2%. This geometric rectification, visualized in Fig. 4(b) and in Fig. 6(d), filters out rotational instabilities.

### 6.4. Ablation study and structural robustness

To isolate the contribution of each algorithmic component, an ablation study is conducted as shown in Table 2.

**Hard projection vs. Lagrangian penalty.** The Soft-$V$ variant, which incorporates $V(\theta)$ as a Lagrangian penalty term in the objective, yields only marginal improvements (76.5% SR). This confirms that soft regularization merely modulates gradient magnitude but lacks the geometric necessity to rectify the update direction. Only the hard analytic projection ($\mathcal{P}$) onto the stability half-space $\mathcal{H}_{\text{stable}}$ ensures that policy updates strictly enter the contractive set.

**Mechanism of adaptive synergy.** The progression from static projection to the full HALO framework underscores the synergy between stability and coherence. While $\mathcal{P}$ provides the convergence certificate, adaptive scheduling ($\eta$)

and alignment $(\cos\phi)$ further refine the trajectory on the joint parameter manifold. HALO suppresses coordination dissonance, as evidenced by the final gap $V$ of 0.09.

*Table 2.* **Ablation matrix on SLH-extreme (pivoting) task.** Results report mean $\pm$ std. $\mathcal{P}$: Lyapunov projection; $\eta$: adaptive scheduling; $\cos\phi$: alignment rectification.

| Variant | $\mathcal{P}$ | $\eta$ | $\cos\phi$ | SR (%) ↑ | Gap $V$ ↓ |
|---|---|---|---|---|---|
| HAPPO baseline | × | × | × | $74.9 \pm 6.0$ | 4.89 |
| Soft-$V$ penalty | × | × | × | $76.5 \pm 5.8$ | 3.21 |
| Static projection | ✓ | × | × | $79.5 \pm 4.8$ | 0.85 |
| HALO w/o align | ✓ | ✓ | × | $80.8 \pm 4.2$ | 0.24 |
| **HALO (full)** | ✓ | ✓ | ✓ | $\mathbf{82.3 \pm 3.8}$ | **0.09** |

### 6.5. Real-world human-robot collaboration

To verify the effectiveness and sim-to-real transferability of HALO for HRC, the real-world evaluation focuses on coordination resilience under partner non-stationarity. We compare HALO against two primary baselines: **PCGrad** and **Robot-Script** (see Appendix B.4).

**Microview resilience analysis.** The effectiveness of HALO

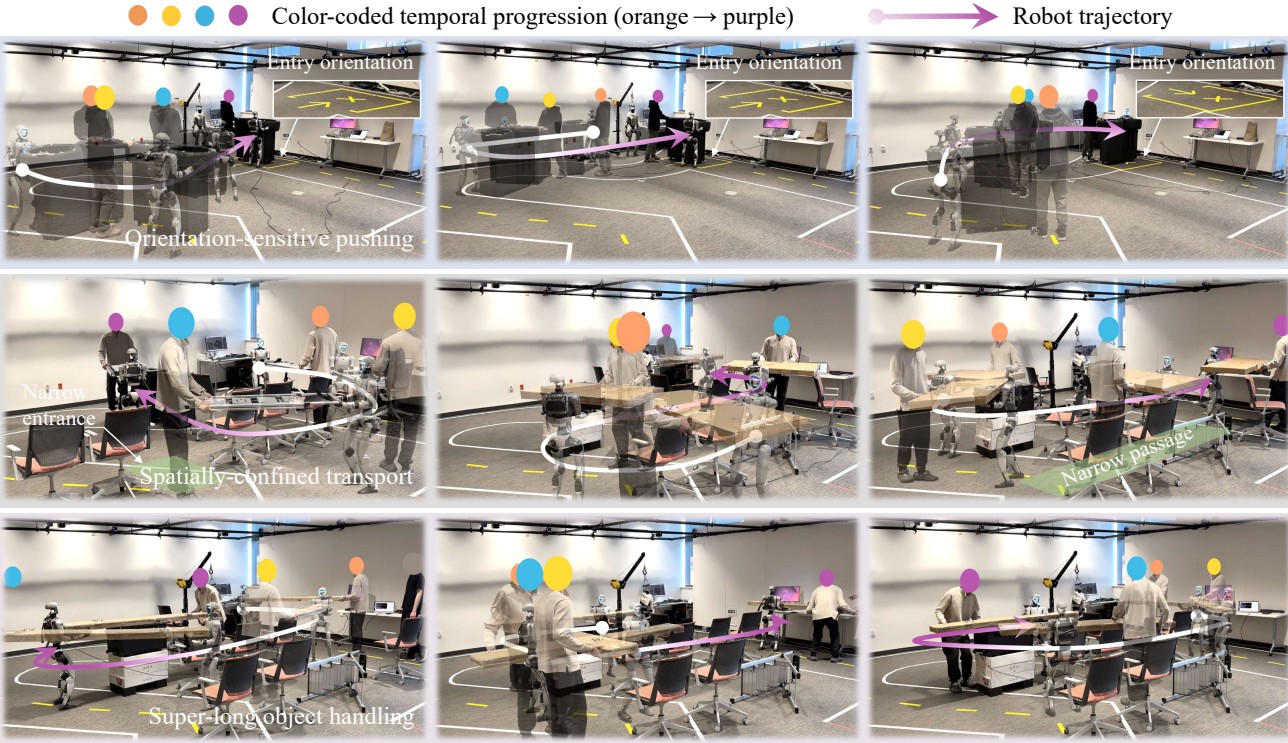

*Figure 8.* Sim-to-real deployment across embodied tasks: macro-view of deployment on OSP (top), SCT (Middle) and SLH (Bottom). Temporal progression is indicated by color gradients; arrows trace the stable trajectories maintained by HALO despite complex physical coupling and human-induced perturbations.

is examined through coordination resilience as shown in Fig. 7. In Fig. 7a, it is observed that the G1 autonomously maintains a horizontal load plane when its partner's height varies. Furthermore, Fig. 7b demonstrates movement synchronization during unscripted 20s obstructions.

*Table 3.* Real-world performance: metrics are mean values over 5 trials. T: time to destination (s); SR: success rate; DR: object drop rate (%); Ang: absolute value of tilt rate ($^{\circ}/s$); $v_d$: post-halt drift (cm/s); Wait: proactive waiting for partner synchrony.

| | Task 1: OSP | | Task 2: SCT | | |
|---|---|---|---|---|---|
| **Method** | **T** ↓ | **SR** ↑ | **T** ↓ | **DR** ↓ | **Ang** ↓ |
| Robot-Script | 74.2 | 100% | 101.6 | 60% | 4.3 |
| PCGrad | 65.2 | 100% | 81.5 | 0% | 2.4 |
| HALO | **61.7** | **100%** | **76.2** | **0%** | **2.2** |

| | Task 3: SLH (Stability Under Halting) | | | | |
|---|---|---|---|---|---|
| **Method** | **Wait** | **T** ↓ | **DR** ↓ | **Ang** ↓ | $v_d$ ↓ |
| Robot-Script | – | – | 100% | 4.9 | – |
| PCGrad | ✓ | 86.9 | 20% | 2.7 | 1.59 |
| HALO | ✓ | **85.6** | **20%** | **2.4** | **1.22** |

**Quantitative analysis.** As synthesized in Table 3, HALO exhibits superior coordination resilience. In OSP and SCT tasks, it significantly reduces time-to-destination (76.2 s)

and minimizes tilt rates ($2.2^{\circ}/s$). Notably, in the SLH task, HALO maintains exceptional stability during unscripted human halting; unlike the robot-script baseline, HALO proactively dissipates residual momentum, yielding a minimal post-halt drift of 1.22 cm/s. This performance underscores HALO's ability to internalize team-level synergy and fluid interaction, as visualized in Fig. 8.

## 7. Conclusion

In this study, we propose HALO to address the inherent structural instabilities in decentralized human-robot collaboration. We establish MARL as a unified paradigm for exploring expansive interaction manifolds, effectively transcending the limitations of traditional scripted human models. We define RG as a variational mismatch between decentralized best-response dynamics and a centralized cooperative ascent direction, reformulating the learning process as a dissipative dynamical system. HALO introduces a formal stability certificate within the policy-parameter manifold, utilizing an optimal quadratic projection to rectify decentralized gradients and ensure the monotonic contraction of coordination disagreement. Our theoretical framework, validated through both large-scale simulations and real-world humanoid deployments, demonstrates that certifying sta-

bility in the parameter space directly translates to superior trajectory coordination and resilience in safety-critical, unstructured environments. Ultimately, HALO provides a foundation for bridging the gap between decentralized individual rationality and global collaborative synergy.

## Software and Data

The project website, which includes videos, additional results, the software, and supplementary materials, is available at: https://HaoZhang-THU.github.io/HALO/.

## Acknowledgements

The authors express their gratitude to the ETAIC Lab (https://ETAIC.github.io/) led by Prof. H. Eric Tseng at UTA, for providing the experimental resources and facilities that made this research possible. We also thank Yisen Li (UPenn), John Song (UTA), and all research assistants and volunteers at ETAIC Lab for their technical support during physical deployments and for helping validate the generalization and robustness of the system.

## Impact Statement

Assistive human–robot collaboration requires robustness to long-tail and out-of-distribution human behaviors that scripted, replay-based, or imitation-driven paradigms cannot adequately capture. These limitations present a practical bottleneck for deploying collaborative robots in settings where non-stationary human intent and physical coupling make failure costly. This work frames HRC as a MARL problem defined over an effectively infinite interaction space, enabling robots to co-adapt with human partners rather than overfit to predefined trajectories. However, decentralized MARL introduces structural instabilities that impede convergence and prevent reliable collaborative behavior. HALO addresses this challenge by introducing a Lyapunov-stabilized learning kernel that contracts coordination disagreement in parameter space.

The resulting stability-centered formulation provides a scalable foundation for the deployment of collaborative robots in industrial workflows, logistics operations, and assistive environments where mixed-autonomy systems must operate in heterogeneous users, dynamic intent patterns, and rare interaction modes. Potential positive societal impacts include reducing physical workload, expanding human operational capacity, and improving safety in labor-intensive settings. Ethical and deployment considerations are primarily related to safety, transparency, and operational responsibility, as well as potential labor displacement and transition of the workforce in society.

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

# A. Detailed theoretical derivations

## A.1. Analytical derivation of the stability gradient

The core of HALO's rectification lies in the gradient of the Lyapunov potential $V(\theta)$. Recall that the rationality gap measures the $L_2$ discrepancy between the independent gradient field $\mathbf{u}_{\text{ind}}$ and the team rationality field $\mathbf{u}_{\text{team}}$ (Zhang et al., 2026; Clempner, 2018):

$$V(\theta) \triangleq \frac{1}{2}\|\mathbf{u}_{\text{ind}}(\theta) - \mathbf{u}_{\text{team}}(\theta)\|_2^2. \tag{12}$$

To compute the stability normal vector $\mathbf{h} \triangleq \nabla_\theta V$, we apply the multivariate chain rule to the inner product. Let $\mathbf{e}(\theta) = \mathbf{u}_{\text{ind}}(\theta) - \mathbf{u}_{\text{team}}(\theta)$ denote the error vector. The gradient is derived as follows (Mesbahi & Egerstedt, 2010):

1. **Vector-valued chain rule:** The gradient $\nabla_\theta V$ is the product of the jacobian of the error field and the error vector itself:

$$\nabla_\theta V = \left(\frac{\partial \mathbf{e}}{\partial \theta}\right)^\top \mathbf{e} = \left(\frac{\partial \mathbf{u}_{\text{ind}}}{\partial \theta} - \frac{\partial \mathbf{u}_{\text{team}}}{\partial \theta}\right)^\top (\mathbf{u}_{\text{ind}} - \mathbf{u}_{\text{team}}). \tag{13}$$

2. **Jacobian components:** We define $\mathbf{H}_{\text{ind}} \in \mathbb{R}^{D \times D}$ as the jacobian of the independent field. In decentralized MARL, this matrix is generally non-symmetric, reflecting the underlying geometry of multi-agent learning dynamics (Leung et al., 2022):

$$\mathbf{H}_{\text{ind}\,jk} = \frac{\partial \mathbf{u}_{\text{ind}\,j}}{\partial \theta_k}. \tag{14}$$

Correspondingly, $\mathbf{H}_{\text{team}} \in \mathbb{R}^{D \times D}$ is the hessian of the global team objective $J(\theta)$ (Gnecco et al., 2012):

$$\mathbf{H}_{\text{team}\,jk} = \frac{\partial^2 J}{\partial \theta_j \partial \theta_k}. \tag{15}$$

3. **Final analytic form:** The stability normal $\mathbf{h}$ is thus:

$$\mathbf{h} = (\mathbf{H}_{\text{ind}} - \mathbf{H}_{\text{team}})^\top (\mathbf{u}_{\text{ind}} - \mathbf{u}_{\text{team}}). \tag{16}$$

## A.2. Explicit derivation of the stability-constrained projection

We assume $V(\theta)$ is $L$-smooth on the parameter manifold $\Theta$. For any two points $\theta, \theta' \in \Theta$, the $L$-smoothness property implies (Nocedal & Wright, 2006):

$$V(\theta') \leq V(\theta) + \langle \nabla V(\theta), \theta' - \theta \rangle + \frac{L}{2}\|\theta' - \theta\|_2^2. \tag{17}$$

Substituting the HALO update law $\theta_{k+1} = \theta_k + \eta \mathbf{d}_k^*$:

$$V(\theta_{k+1}) \leq V(\theta_k) + \eta\langle \nabla_\theta V(\theta_k), \mathbf{d}_k^* \rangle + \frac{L\eta^2}{2}\|\mathbf{d}_k^*\|_2^2 \tag{18}$$

$$= V(\theta_k) + \eta\langle \mathbf{h}, \mathbf{d}_k^* \rangle + \frac{L\eta^2}{2}\|\mathbf{d}_k^*\|_2^2. \tag{19}$$

To ensure the monotonic dissipation of the rationality gap $V(\theta)$, HALO formulates a quadratic programming problem (Duchi et al., 2008):

$$\mathbf{d}^* = \arg\min_{\mathbf{d} \in \mathbb{R}^D} \frac{1}{2}\|\mathbf{d} - \mathbf{u}_{\text{ind}}\|_2^2$$
$$\text{s.t.} \quad \langle \nabla_\theta V, \mathbf{d} \rangle \leq -\sigma V, \tag{20}$$

We define the Lagrangian function $\mathcal{L}(\mathbf{d}, \lambda)$ with a multiplier $\lambda \geq 0$ (Fletcher, 2013):

$$\mathcal{L}(\mathbf{d}, \lambda) = \frac{1}{2}\|\mathbf{d} - \mathbf{u}_{\text{ind}}\|_2^2 + \lambda(\mathbf{h}^\top \mathbf{d} + \sigma V). \tag{21}$$

The KKT stationarity condition $\nabla_{\mathbf{d}}\mathcal{L} = 0$ yields (Hempel et al., 2017):

$$\mathbf{d}^* = \mathbf{u}_{\text{ind}} - \lambda\mathbf{h}. \tag{22}$$

Solving $\lambda(\mathbf{h}^\top\mathbf{d}^* + \sigma V) = 0$:

1. Case 1: If $\mathbf{h}^\top\mathbf{u}_{\text{ind}} + \sigma V \leq 0$, then $\lambda^* = 0$.

2. Case 2: If $\mathbf{h}^\top\mathbf{u}_{\text{ind}} + \sigma V > 0$, the constraint is active. Substituting $\mathbf{d}^*$ into the boundary:

$$\mathbf{h}^\top(\mathbf{u}_{\text{ind}} - \lambda^*\mathbf{h}) + \sigma V = 0 \implies \lambda^* = \frac{\mathbf{h}^\top\mathbf{u}_{\text{ind}} + \sigma V}{\|\mathbf{h}\|_2^2}. \tag{23}$$

The unified closed-form solution is:

$$\lambda^* = \max\left(0, \frac{\langle\mathbf{h}, \mathbf{u}_{\text{ind}}\rangle + \sigma V}{\|\mathbf{h}\|_2^2 + \epsilon}\right), \quad \mathbf{d}^* = \mathbf{u}_{\text{ind}} - \lambda^*\mathbf{h}. \tag{24}$$

### A.3. Asymptotic convergence analysis

From the descent inequality $V(\theta_{k+1}) \leq V(\theta_k) - \eta_k\sigma V(\theta_k) + \frac{L\eta_k^2}{2}\|\mathbf{d}_k^*\|_2^2$, summing from $k = 0$ to $K$:

$$\sigma\sum_{k=0}^{K}\eta_k V(\theta_k) \leq V(\theta_0) - V(\theta_{K+1}) + \frac{L}{2}\sum_{k=0}^{K}\eta_k^2\|\mathbf{d}_k^*\|_2^2. \tag{25}$$

Under Robbins-Monro conditions ($\sum\eta_k = \infty, \sum\eta_k^2 < \infty$) and bounded gradients $\|\mathbf{d}^*\| \leq G$ (Chen et al., 2026; Bottou et al., 2018):

$$\sigma\sum_{k=0}^{\infty}\eta_k V(\theta_k) \leq V(\theta_0) + \frac{LG^2}{2}\sum_{k=0}^{\infty}\eta_k^2 < \infty. \tag{26}$$

Since $\sum\eta_k$ diverges, it must hold that $\liminf_{k\to\infty}V(\theta_k) = 0$. Due to the monotonicity and uniform continuity of the update, we conclude:

$$\lim_{k\to\infty}V(\theta_k) = 0 \implies \lim_{k\to\infty}\|\mathbf{u}_{\text{ind}}(\theta_k) - \mathbf{u}_{\text{team}}(\theta_k)\|_2 = 0. \tag{27}$$

This confirms that HALO asymptotically collapses the learning dynamics onto the rationality agreement manifold.

## B. Implementation and experimentation details

### B.1. Technical details of the hierarchical control architecture

The coordination framework operates via a tri-level control hierarchy designed to decouple long-term mission guidance from high-frequency physical stabilization (Zhang et al., 2025). Each layer operates at a distinct temporal scale and functional scope as specified in table 4.

**Top-level global mission planner.** At the start of each episode, the system establishes a geometric backbone using the path planning algorithm, which can also be realized alternatively using vision language model (VLM). The A* or VLM planner generates a collision-free path for the object's center of mass (CoM) based on a static environmental map. This path serves as the reference trajectory for the downstream tactical and execution layers.

**Mid-level tactical MARL policy.** Operating at 2 Hz (500ms intervals), the mid-level MARL policy acts as the coordination command generator. It receives spatially-sampled waypoints from the global path and generates motion command for whole-body controller (WBC).

**Bottom-level whole-body controller.** The bottom-level execution is handled by a WBC policy operating at 50 Hz (20ms intervals). This layer ensures the dynamic stability of the G1 humanoid robot by tracking the 11-dimensional (11D) commands generated by the MARL policy. To bridge the frequency gap between the MARL (2 Hz) and WBC (50 Hz) layers, mid-level commands are held constant via a zero-order hold mechanism within each 500ms decision cycle.

*Table 4.* **Temporal and functional specification of the control hierarchy.**

| Layer | Frequency | Update schedule | Core functional scope |
|---|---|---|---|
| Top-level | N/A | Episode initialization | Global path planning (A* or VLM) for object CoM |
| Mid-level | 2 Hz | Tactical MARL policy | Motion command generation for collaboration |
| Bottom-level | 50 Hz | WBC execution | Body balancing and motion command tracking |

## B.2. Observation and command space formulations

To facilitate long-horizon navigation without the computational overhead of recurrent architectures, we implement a dual-snapshot temporal encoding alongside a spatially-sampled look-ahead mechanism. Each agent processes a 210-dimensional observation vector $\mathbf{o}_i$, detailed in table 5. The strategic guidance is provided by a sliding window of waypoints $\{p_k\}_{k=1}^5$ extracted from the A* or VLM-planned global path at fixed curvilinear intervals (1m to 5m) relative to the robot's current position. All exteroceptive features—including waypoints, partner relative pose, and object geometry—are transformed into the agent's egocentric local frame to ensure spatial invariance.

*Table 5.* **Composition of the 210-dimensional observation space.**

| Feature domain | Dim | Description (Egocentric coordinates) |
|---|---|---|
| Look-ahead guidance | 10 | 2D XY waypoints sampled via a sliding window along the global path |
| Self-proprioception | 13 | XY pos/vel, yaw, CoM height, torso pitch, wrist-to-shoulder XYZ |
| Partner observation | 13 | Relative XY pos/vel, yaw, CoM height, torso pitch, wrist-to-shoulder XYZ |
| Object geometry | 18 | 4 Top-surface corners XYZ, CoM XYZ pos, CoM XYZ vel |
| Contact feedback | 4 | Contact signals (left/right end-effector contact for both of the agents) |
| Temporal features | 174 | Snapshot accumulation (58D per frame $\times$ 3 snapshots) |
| Environment awareness | 36 | 36-ray synthetic proximity, normalized as $1 - d/d_{max}$ |
| **Total observation** | **210** | **Input to the 2 Hz tactical MARL policy** |

The command space utilizes a delta-over-base mechanism for precise end-effector coordination. As shown in table 6, the MARL policy modulates a 3D spatial offset ($\Delta \mathbf{p}$) superimposed onto a task-specific base pose.

*Table 6.* **MARL command space specification (11D).**

| Command set | Dim | Formulation | Physical meaning |
|---|---|---|---|
| Locomotion base | 3 | Absolute | Target velocities $[v_x, v_y]$ and orientation (yaw angle) |
| Postural setpoints | 2 | Absolute | Target CoM height ($H_{\text{CoM}}$) and torso pitch ($\alpha_{\text{torso}}$) |
| Wrist modulation | 6 | Relative delta | 3D XYZ offset added to task-specific base poses for both Wrist |

## B.3. Hyperparameters and training configuration

The tactical coordinator is implemented via a shared-parameter HAPPO framework. This architecture facilitates CTDE, effectively mitigating the non-stationarity inherent in multi-agent coordination. Both actor and critic networks utilize a multilayer perceptron (MLP) backbone with hidden layers of [256, 256, 128]. To ensure stable gradient propagation over the $2.0 \times 10^9$ steps, we employ orthogonal initialization with a gain of $\sqrt{2}$. The optimization process utilizes the Adam optimizer coupled with a cosine annealing learning rate schedule, balancing exploratory breadth with asymptotic convergence. Detailed hyperparameters are synthesized in table 7.

The training objective is defined through a path-wise reward that prioritizes geodesic progress along the A* or VLM pre-planned object CoM movement trajectory while enforcing structural stability. This formulation provides a dense reward signal that guides agents through non-convex environmental constraints by focusing on incremental advancement.

*Table 7.* **Optimization and topology hyperparameters.**

| Optimization parameter | Value | Architecture parameter | Value |
|---|---|---|---|
| Learning rate ($\alpha$) | $1.0 \times 10^{-4}$ | Hidden layers (actor/critic) | [256, 256, 128] |
| Epochs per update | 10 | Activation function | ReLU |
| Minibatch count | 16 | Layer initialization | Orthogonal |
| Entropy coefficient | 0.01 | Optimizer | Adam |
| Discount factor ($\gamma$) | 0.99 | Weight decay ($L_2$) | $1.0 \times 10^{-4}$ |
| GAE parameter ($\lambda$) | 0.95 | Gradient clipping ($\|g\|_2$) | 10.0 |
| Clipping epsilon ($\epsilon$) | 0.2 | Learning rate schedule | Cosine annealing |
| Value loss coefficient | 0.5 | Total MARL action steps | $2.0 \times 10^9$ |

## B.4. Baseline robot-script implementation

To evaluate the necessity of the MARL framework and the coordination gains of HALO, we implement a robot-script baseline based on independent PPO (IPPO). This baseline simplifies the HRC into a single-agent control task by modeling the human partner as a stochastic dynamical load source with predefined mobility. The robot-script policy utilizes a MLP backbone with hidden layers of [256, 256, 128], identical to the architecture employed in HALO. To ensure a fair yet rigorous comparison, the action and observation spaces are aligned with those of HALO, with the exception of collaborative features. Specifically, the state of the interactive partner is replaced by the simplified kinematic state of the human proxy (represented as two boxes), which comprises the relative position, velocity, and orientation.

*Table 8.* Stochastic perturbations for simulating human-like dynamical loads.

| Noise type | Distribution | Magnitude | Target simulation phenomenon |
|---|---|---|---|
| Vertical jitter | Gaussian | 2.0 cm | Gait-induced oscillations and lifting shifts |
| Velocity noise | Uniform | 0.1 m/s | Non-uniform walking speed and intent changes |
| Yaw perturbation | Gaussian | 3.0 deg | Subtle directional adjustments during transport |

*Table 9.* Robot-script training hyperparameters.

| Optimization parameter | Value | Architecture parameter | Value |
|---|---|---|---|
| Learning rate ($\alpha$) | $1.0 \times 10^{-4}$ | Hidden layers (actor/critic) | [256, 256, 128] |
| Epochs per update | 10 | Activation function | ReLU |
| Minibatch count | 16 | Layer initialization | Orthogonal |
| Entropy coefficient | 0.01 | Optimizer | Adam |
| Discount factor ($\gamma$) | 0.99 | Weight decay ($L_2$) | $1.0 \times 10^{-4}$ |
| Learning rate schedule | Cosine annealing | Total RL action steps | $2.0 \times 10^9$ |

The policy is trained for $2.0 \times 10^9$ steps, using the IPPO algorithm. We employ orthogonal initialization and the Adam optimizer with a cosine annealing learning rate schedule to maintain consistency with our HALO framework. To simulate gait-induced oscillations and intentional shifts, multi-axis stochastic noise is injected into the human proxy's motion. These perturbations, detailed in Table 8, force the robot to implicitly compensate for interaction residuals through individual proprioceptive feedback. The comprehensive training and optimization hyperparameters are provided in Table 9.

