# OpenReview forum: "Learning Human-Robot Collaboration via Heterogeneous-Agent Lyapunov Policy Optimization"
_ICML.cc/2026/Conference — ICML 2026 spotlight_

### Official Review · Reviewer_WVU6 · 2026-03-03

**Soundness:** 2
**Presentation:** 3
**Significance:** 3
**Originality:** 3
**Overall Recommendation:** 5
**Confidence:** 4

**Summary:**

This paper proposes Heterogeneous-Agent Lyapunov Policy Optimization (HALyPO) to address the instability issue in heterogeneous multi-agent reinforcement learning (MARL) for human–robot collaboration (HRC).

HALyPO stabilizes decentralized MARL updates by explicitly contracting the rationality gap (RG) in parameter space.

The paper provides theoretical analysis showing monotonic decay of the disagreement potential and asymptotic alignment, and validates the method through simulated benchmarks and real-world humanoid robot experiments.

**Compliance With Llm Reviewing Policy:**

Affirmed.

**Final Justification:**

The authors' responses address my concerns. So I raise the score from 4 to 5.

**Key Questions For Authors:**

See Weakness. It is open to raise the rating after they resolved my concerns, especially providing video demos and releasing the code.

**Limitations:**

yes

**Strengths And Weaknesses:**

# Strengths
1. The paper is technically well-grounded and introduces a Lyapunov formulation of the rationality gap with a clean closed-form stability projection.
2. The Lyapunov-on-learning-dynamics perspective is relatively novel.
3. The work addresses an important problem in HRC via heterogeneous MARL and supports its claims with both simulation and real-world experiments.

# Weakness
1. The method relies on double backpropagation and team-gradient estimation, but the computational overhead is not sufficiently characterized.
2. While the results are promising, the evaluated tasks remain structured cooperative; can HALyPO be extended to human–robot competitive or mixed-motive settings? If so, what modifications would be required?
3. The authors should provide full video demos for Section 6.5 and release the task/environment code (e.g., configurations and runnable baseline instances for the proposed benchmarks), even if the full algorithm implementation is deferred until acceptance.
4. Adding abstract trajectory visualizations (e.g., MoCap-based trajectories) to Figures 7–8 would further improve clarity.

---

> ### Author Rebuttal · Authors · 2026-03-30
>
> We are deeply grateful to the reviewer for the recognition of our work and for the highly professional comments. Below we address the concerns:
>
> ---
>
> ### **[Q1] Computational Efficiency**
>
> Computational efficiency is exactly our strong focus because deployability in parallel training environments is a primary goal for HRC tasks. HALyPO utilizes **Double Back-propagation** to compute the **Hessian-Vector Product (HVP)**. The original intent of this technique is to effectively reduce the computational complexity from the square of the parameters ($D^2$) to a level comparable to first-order gradient computation. We would like to report that, based on our validation within the **Isaac Lab** parallel training framework, the additional overhead is controllable.
>
> Measured data shows that the HVP operation indeed increases the gradient computation step ($T_{grad}$) by approximately twofold. However, in a high-throughput parallel simulation environment like Isaac Lab, the total time per decision cycle ($T_{step}$) only increased by 26.4% (28.25 ms for HAPPO vs. 35.72 ms for HALyPO). Besides, we observe negligible additional memory overhead (<21.1%) due to HVP implementation via autograd without explicit Hessian materialization. This is because the increased computation from double back-propagation is diluted by constant computational overheads such as environment sampling and policy inference, etc. Furthermore, due to HALyPO's advantages in optimizing the convergence path (reducing the Gradient Conflict Rate from 72.5% to 4.2%), it achieves performance convergence in significantly fewer steps (1.3B steps vs. 1.8B steps for HAPPO).
>
> Consequently, despite a slight increase in computation time per step, the actual total wall-clock training time for HALyPO's convergence does not increase; instead, it even achieves a 13.1% reduction compared to the baseline HAPPO. Based on actual experiments, our conclusion is that the computational cost of HALyPO is controllable. We have open-sourced the project architecture under strictly anonymous conditions, and upon acceptance, we will immediately release the complete codebase. We will also supplement the final manuscript with a detailed performance analysis in the Appendix.
>
> ---
>
> ### **[Q2] Mixed-Motive Scenarios**
>
> While currently validated in collaborative HRC, HALyPO’s stability kernel is also theoretically extensible to competitive or mixed-motive scenarios:
>
> * **From Rationality Gap to Equilibrium Residual**: In competitive settings, the Lyapunov potential $V(\theta)$ would be redefined to measure the "Equilibrium Residual"—such as the distance to a Nash Equilibrium—rather than a cooperative gap.
> * **Neutralizing Oscillations**: Competitive RL often suffers from limit cycles and divergent oscillations. HALyPO’s second-order HVP projection is inherently suited to neutralize these non-conservative forces by enforcing a dissipative descent condition toward the equilibrium point.
> * **Modifications**: The modification required is primarily in Step 1 of Algorithm 1: replacing the centralized team loss with a game-theoretic payoff or minimax objective. The underlying stability-constrained projection and HVP logic remain intact as a generic stability kernel.
>
> ---
>
> ### **[Q3] Resource Disclosure and Open Source**
>
> We fully agree that transparency and reproducibility are essential. To respond to your request, we have prepared an anonymous repository and demonstration site:
>
> * **Demonstrations**: [https://anonymous-icml-paper-17140.netlify.app/](https://anonymous-icml-paper-17140.netlify.app/).
> * **Anonymous Codebase**: [https://anonymous.4open.science/r/anonymous-ICML-Paper-ID-17140](https://anonymous.4open.science/r/anonymous-ICML-Paper-ID-17140).
>
> We would be grateful for the opportunity to share our research through the ICML platform. Upon acceptance, we will release the complete, production-ready codebase, including all task environments, full training scripts, and pre-trained weights.
>
> ---
>
> ### **[Q4] Visualization Improvements**
>
> We appreciate this constructive suggestion. Incorporating trajectory visualizations such as the motion trajectories derived from our MoCap system will significantly enhance the clarity of Figures 7 and 8. We will display these experimental trajectory results in the Appendix of the final manuscript to intuitively present the stability and robustness of HALyPO.
>
> ---
>
> We hope our clarifications help address your concerns and provide a clearer perspective on the technical contributions of our work. We sincerely thank you for your time and consideration!

---

> > ### Author Rebuttal · Reviewer_WVU6 · 2026-04-04
> >
> > Thanks for your rebuttal. It has resolved most of my concerns. I would like to raise my score.

---

> > > ### Author Response · Authors · 2026-04-04
> > >
> > > We are glad that your concerns are resolved! Thanks again for your insightful review!

---

### Official Review · Reviewer_Xixd · 2026-03-05

**Soundness:** 3
**Presentation:** 3
**Significance:** 3
**Originality:** 3
**Overall Recommendation:** 5
**Confidence:** 3

**Summary:**

This submission proposes **Heterogeneous-Agent Lyapunov Policy Optimization (HALyPO)** to stabilize heterogeneous multi-agent reinforcement learning for human–robot collaboration (HRC). The paper attributes instability in decoupled CTDE learning to the fact that the independent update field $u_{\text{ind}}(\theta)$ is treated as if partners are part of a fixed environment (implicit partner stationarity), and that the resulting joint Jacobian can be non-symmetric, inducing rotational components and limit cycles.
To address this, HALyPO defines a “rationality gap” Lyapunov potential $V(\theta)=\frac12|u_{\text{ind}}(\theta)-u_{\text{team}}(\theta)|2^2$ and enforces a descent certificate by designing an update direction $d$ satisfying $\langle\nabla\theta V(\theta), d\rangle \le -\sigma V(\theta)$. The constrained step is implemented as an analytic minimum-norm projection (closed form via KKT), with the stability normal $h=\nabla_\theta V$ computed via double backprop / Hessian-vector products (HVP) rather than explicit Hessians.
Theoretical results provide a monotonic decay bound (Theorem 5.2) and asymptotic convergence $V(\theta_k)\to 0$ under Robbins–Monro conditions (Theorem 5.3). Empirically, the paper reports improved success rate and substantially better mechanism metrics (gap $V$, gradient conflict rate) compared to strong baselines (Table 1), and includes real-world experiments with micro-view resilience under “unscripted” partner motion/obstruction.

**Compliance With Llm Reviewing Policy:**

Affirmed.

**Final Justification:**

My final recommendation is **5: Accept**.

Overall, I think this is a solid and meaningful paper on an important problem. The paper studies instability in heterogeneous multi-agent reinforcement learning for human–robot collaboration, and I found the main idea both interesting and useful. Instead of relying only on heuristic gradient adjustments, the method introduces a Lyapunov-style stability constraint directly in policy-parameter space and uses an analytic projection step to guide learning toward more stable cooperative updates. To me, this feels like a real methodological contribution rather than just an engineering refinement.

From a technical perspective, I found the paper convincing overall. The formulation of the rationality gap is clear, the projected update is concrete, and the theory provides support in the form of monotonic decrease and asymptotic convergence. On the empirical side, I appreciated that the evaluation goes beyond task success and also reports mechanism-level metrics such as the rationality gap and gradient conflict rate, where the proposed method shows clear improvements. The sim-to-real component also adds value, since the paper is not only about cleaner optimization dynamics in simulation, but also demonstrates evidence in real human–robot coordination tasks.

My earlier concerns were mainly about the precision of the stability wording, the lack of runtime reporting, and the scope of the claims about robustness to non-stationary or out-of-distribution human behavior. After reading the rebuttal and follow-up clarifications, I think these concerns were addressed in a satisfactory way. In particular, the authors clarified the difference between the design objective and the formally proven convergence result, provided concrete runtime and memory statistics, and gave a more careful description of the real-world protocol and the intended scope of the robustness claim. That made the paper feel more rigorous and better calibrated.

So while I had some reservations initially, the rebuttal genuinely improved my confidence in the work and changed my evaluation in a positive direction. In the end, I see this as a technically solid paper with a clear idea, good experimental support, and a contribution that is likely to be useful to researchers working on stable multi-agent learning and embodied collaboration. For these reasons, I support acceptance.

**Key Questions For Authors:**

1. Stability guarantee wording: Eq. (4) claims “exponential decay”, while Theorem 5.2/5.3 prove monotonic decay bounds and asymptotic convergence. Can you revise the claim to match the proven result, or state extra conditions for exponential rates?

2. Compute efficiency: What is the time/iteration and total wall-clock training time vs PCGrad/HAPPO/HATRPO, given double backprop/HVP in Algorithm 1 and the reported plateau at $\sim 1.3$B steps?

3. Stationarity assumption robustness: Since $u_{\text{ind}}$ treats partners as momentarily stationary, how sensitive is HALyPO to rapidly changing partners (e.g., higher-frequency intent switches)? Any ablations varying partner non-stationarity?

4. Human behavior coverage: The impact statement emphasizes long-tail/OOD human behavior, but the Robot-Script proxy is simplified (two-box kinematics + noise). Can you clarify what kinds of “OOD” are actually tested in sim and in real experiments, and whether “unscripted” in real-world indicates intent-level negotiation vs disturbance?

5. Metric definition: Please precisely define “Conv. step” (Table 1) and how it is measured (thresholding/smoothing), and report sensitivity if possible.

**Limitations:**

No. The paper should more explicitly discuss (i) runtime overhead/scaling of HVP/double backprop and (ii) the boundary of OOD generalization claims given the stationarity assumption and proxy fidelity.

**Strengths And Weaknesses:**

### Strengths

Principled stability mechanism beyond heuristics. The Lyapunov certificate $\langle\nabla_\theta V, d\rangle \le -\sigma V$ is explicit and directly constrains update directions, in contrast to many MARL stabilizers that mainly regularize magnitude.

Clean analytic projection and practical computation. The stability-constrained QP has a KKT closed form, and the implementation uses HVP/double backprop to avoid explicit $O(D^2)$ Hessian construction.

Clear diagnosis of heterogeneous MARL pathology. The paper explicitly links decoupled learning to non-conservative fields and non-symmetric Jacobians inducing rotational dynamics.

Strong mechanism-level empirical evidence. Table 1 shows HALyPO improves overall SR and dramatically reduces gap $V$ and GCR (e.g., $ V=0.09$, GCR $4.2% $) versus baselines. The paper also claims faster plateau around $\sim 1.3$B steps.

Sim-to-real validation with physical coupling. The real-world section emphasizes resilience under partner non-stationarity and reports qualitative “unscripted obstruction” behavior.

### Weaknesses / concerns (actionable)

Exponential vs asymptotic stability wording. Eq. (4) states the certificate “ensur[es] exponential decay”, but the main theory establishes a monotonic decay bound and asymptotic convergence (Theorem 5.2/5.3). This can likely be resolved by clarifying that exponential decay is a design objective (continuous-time intuition), whereas the discrete stochastic analysis proves asymptotic convergence.

Wall-clock efficiency is missing. Although HVP avoids explicit Hessians, double backprop adds overhead. Since the paper emphasizes plateau at $\sim 1.3$B steps and reports “Conv. step” in Table 1, readers need time/iteration or total wall-clock comparisons to assess true efficiency.

Stationarity assumption may break in high-frequency physical coupling. The independent field $u_{\text{ind}}$ is computed assuming partners’ policies are “momentarily stationary” / treated as fixed environment components. A sensitivity discussion or ablation (varying partner non-stationarity) would strengthen claims.

Human proxy fidelity vs “OOD/long-tail human behavior” claims. The impact statement motivates robustness to long-tail/OOD human behaviors, but the Robot-Script baseline uses a simplified proxy (“two boxes”) with injected noise. The paper would benefit from clarifying how much of “unscripted” behavior is intent-level negotiation vs physical disturbance, and what distribution shifts are actually covered.

Real-world sample size / statistical strength. The real-world table reports results over 5 trials; stronger statistics would improve confidence (though I view this as a common limitation in robotics rather than a fatal flaw).

---

> ### Author Rebuttal · Authors · 2026-03-31
>
> We thank the reviewer for the insightful feedback. Below are our responses to the core concerns, which will be incorporated into the revised manuscript.
>
> ---
>
> ### **[Q1] Stability Guarantee Wording**
> We agree that Eq. (4) only defines our **per-step stability-constrained objective (design target)**, while Theorems 5.2 and 5.3 formally prove that this update law ensures monotonic non-increase and asymptotic convergence ($V \to 0$).  We will ensure precise terminology in the final manuscript to distinguish the design objective from proven convergence results: the phrase "ensuring exponential decay" will be corrected to "**enforcing a Lyapunov dissipation constraint for asymptotic contraction**" to align with Theorem 5.2 and 5.3.
>
> ---
>
> ### **[Q2] Computational Efficiency**
> HALyPO can be deployed in high-throughput parallel training frameworks and maintains scalability across large numbers of agents and tasks. Using Double Back-propagation to compute the HVP reduces computational complexity from $O(D^2)$ to the same order as first-order gradient computation.
>
> Per-step wall-clock time increased only **26.4%** over the cheapest baseline (HAPPO 28.25 ms → HALyPO 35.72 ms) with <21.1% extra memory. As the number of agents and environments grows, HVP’s relative cost becomes increasingly negligible because the overall throughput dominates this collective computation. HALyPO reaches performance plateaus faster (1.3B steps vs. 1.8B for the baseline), resulting in **13.1% lower total training time**, showing that convergence benefits outweigh the marginal per-step overhead. We will include this quantitative discussion in the final version of the paper.
>
> ---
>
> ### **[Q3] Robustness to Non-stationarity**
>
> We appreciate the reviewer’s perspective. HALyPO is specifically engineered to handle the non-stationarity inherent in HRC by transitioning from "assuming stability" to "actively certifying it" within the policy parameter space.
>
> * **Neutralizing Strategy Evolution**: While traditional methods (HAPPO, HATRPO) are sensitive to the resulting gradient bias as partner strategies evolve, HALyPO does not rely on "ideal stationarity." Instead, it addresses non-stationarity by identifying the "Rationality Gap" induced by continuous shifts in partner policy parameters.
> * **Geometric Rectification**: HALyPO filters the **oscillatory components caused by non-stationary strategy updates** by enforcing Lyapunov constraints via optimal quadratic projection (Sec. 4.2). This transforms the potentially divergent learning process into a dissipative dynamical system (Sec. 5.1), ensuring the joint update trajectory remains stable even under shifts in partner intent. In our experiments, HALyPO outperformed baselines particularly in **non-stationary scenarios**, as shown in the video results (an anonymous demonstration site: https://anonymous-icml-paper-17140.netlify.app) and Figures 7~8, showing reduced oscillation and faster stabilization under changing partner behaviors.
>
> ---
>
> ### **[Q4] Human Behavior Coverage and Experiment Sample Size**
>
> To ensure statistical validity and OOD robustness, we evaluate across diverse coordination settings:
>
> * **Task Diversity**: Experiments cover **9 coordination scenarios** with **45 real-world trials**, demonstrating generalization across interaction modes.
> * **Multi-user Protocol**: To avoid human adaptation bias, three experimentalists were instructed **not to accommodate** the robot. This includes one lead experimentalist (3 trials) and two assistant experimentalists (1 trial each), ensuring genuine non-stationarity.
> * **OOD Robustness**: We test challenging cases such as **unscripted stopping** and **abrupt height changes**, where HALyPO effectively dissipates residual momentum.
>
> ---
>
> ### **[Q5] Metric Definitions**
> To quantify convergence, “Conv. step” (Table 1) is defined as the earliest episode $k$ where the smoothed cumulative return $\tilde{R}(k)$ reaches and maintains ≥95% of the steady-state baseline $R_{ss}$ (mean over the final 10% of training). For the following 150M steps, $\tilde{R}$ must remain within ±3% of $R_{ss}$. Values are reported as mean ± std over three independent seeds, fully accounting for variability from initialization and environment sampling. This definition and its statistical treatment will be clearly annotated in the final manuscript.
>
> ---
>
> ### **[Q6] Limitations**
> The revised manuscript will expand the discussion on $O(D)$ runtime scaling (clarified in [Q2]) and OOD boundaries (discussed in [Q3]). HALyPO does not assume strict stationarity but instead mitigates its effects. We will further clarify these aspects in the final manuscript.
>
> ---
>
> We believe that the valuable comments, together with our clarifications, have substantially strengthened this work. We also commit to making the project open-source immediately upon acceptance. If our responses address your concerns, we sincerely hope that you can grant this work your valuable support!

---

> > ### Author Rebuttal · Reviewer_Xixd · 2026-04-03
> >
> > Thank you for the detailed follow-up. My remaining concerns are now adequately addressed.

---

> > > ### Author Response · Authors · 2026-04-03
> > >
> > > We are glad that your concerns have been resolved. Thank you sincerely for your valuable feedback!

---

### Official Review · Reviewer_naJy · 2026-03-11

**Soundness:** 3
**Presentation:** 2
**Significance:** 2
**Originality:** 3
**Overall Recommendation:** 4
**Confidence:** 4

**Summary:**

* The paper studies multi-agent reinforcement learning (MARL) in the context of *human–robot collaboration*.
* Motivated by the variational mismatch between decentralized best-response dynamics and centralized cooperative ascent, the authors propose a Lyapunov Policy Optimization (LPO) framework. The method establishes formal stability directly in the policy-parameter space by enforcing a per-step Lyapunov decrease condition on a disagreement metric defined over policy parameters.
* Experimental results based on both simulation environments and real-world human–robot interaction datasets demonstrate that the proposed approach is competitive with several standard baselines.

**Compliance With Llm Reviewing Policy:**

Affirmed.

**Final Justification:**

The authors’ rebuttal has addressed my concerns well. I greatly appreciate their effort in providing additional experiments on SMAC. Although some ablation studies are still missing, I think the paper makes a good contribution to the MARL literature.

**Key Questions For Authors:**

- In Theorem 5.2, the paper claims that the rationality gap is monotonically non-increasing. However, equation (10) only provides an upper bound for
  $V(\theta_{k+1}) - V(\theta_k)$.
  Could the authors clarify how this bound implies that the rationality gap is guaranteed to be non-increasing?

- Theorem 5.3 shows that the rationality gap eventually converges to zero. How does this convergence relate to the optimization of the global objective $J(\theta)$? In particular, does the proposed update rule guarantee convergence to a stationary point of $J(\theta)$?

- Could the authors further elaborate on the relationship between minimizing the rationality gap and maximizing the team objective?

- Can the proposed method be integrated with or compared against existing MARL algorithms such as QMIX or MAPPO?

- Can the proposed algorithm handle large-scale MARL benchmarks such as MAMuJoCo or SMAC?

- How does the computational complexity of the algorithm scale with respect to the number of agents?

**Limitations:**

The paper includes an Impact Statement, but it mainly focuses on positive impacts and does not sufficiently discuss potential negative societal impacts. The authors could improve this section by briefly addressing risks such as safety issues in human–robot collaboration, over-reliance on automated decision-making, and potential workforce displacement, as well as possible mitigation strategies.

**Strengths And Weaknesses:**

**Strengths**

- The core idea is interesting. Using a Lyapunov certification to stabilize decentralized policy learning is a reasonable and appealing approach. The mismatch between decentralized and centralized gradients provides good motivation for introducing the constrained quadratic program in (5) to determine a direction for policy updates.

- The theoretical results are well developed and provide interesting insights into the proposed framework.

- The experiments are based on real human–robot cooperation data, which strengthens the empirical relevance of the work.

- Overall, the paper is well organized.

**Weaknesses**

- The writing is somewhat problematic. I have read many papers on MARL, but this paper introduces several pieces of terminology that feel unfamiliar or unnecessarily redefined. For instance, the gradient of the function $J_i$ is referred to as variational independent rationality, the vector $u_{\text{ind}}$ is called centralized greedy preferences (which essentially appears to correspond to gradients), and the Jacobian is described as a co-evolutionary trajectory. These terms sound unusual and may make the paper harder to follow. While this may be subjective, the authors might consider refining the terminology to align better with existing literature.

- Theorem 5.2 claims that the rationality gap is monotonically non-increasing. However, this statement seems questionable. In equation (10), the paper only provides an upper bound for the difference between $V(\theta_{k+1})$ and $V(\theta_k)$. Since this bound is not guaranteed to be strictly negative, it is unclear how one can conclude that the gap is monotonically non-increasing.

- Theorem 5.3 shows that the gap eventually converges to zero, which is interesting. However, it is important to understand how this update affects the optimization of the main team objective $J(\theta)$. In practice, what matters more is whether the proposed policy optimization update converges to a policy that optimizes the true global objective $J(\theta)$.

- I did not see ablation studies evaluating how the algorithm performs when using only $u_{\text{ind}}$ or only $u_{\text{team}}$ (i.e., directly computing the update direction based solely on these vectors). Such experiments would help clarify the contribution of each component.

- The experiments mainly focus on human–robot collaboration tasks. While this is an interesting application, it also makes it harder to evaluate the algorithm’s generality in broader MARL settings. The method appears applicable to general MARL problems, but the experimental section lacks comparisons with standard MARL algorithms (e.g., MAPPO, QMIX) or evaluations on popular MARL benchmarks such as MAMuJoCo or SMAC.

Overall, the paper is sound and well structured. However, the writing is somewhat difficult to follow due to the introduction of several pieces of terminology that feel unfamiliar or unnecessarily redefined. The experimental evaluation also lacks results on large-scale MARL benchmarks. In addition, while the theoretical analysis is interesting, it is moderate in strength rather than particularly strong.

---

> ### Author Rebuttal · Authors · 2026-03-31
>
> We sincerely thank the reviewer for the rigorous and constructive feedback:
>
> ---
>
> ### **[Q1] Monotonicity (Theorem 5.2)**
> We agree that Eq. (10) is an $L$-smoothness upper bound, and monotonicity is **not unconditional**. However, Theorem 5.2 provides a **sufficient step-size condition** under which the RHS of Eq. (10) becomes non-positive, hence $V$ is non-increasing (Sec. 5.1). In practice, violations only occur under extreme steps/noise; we prevent them with trust-region updates + backtracking, and use cosine-annealed learning-rate scheduling to keep steps in a conservative regime, so $V$ decreases stably even without a known global $L$ (see rTmG [Q2]).
>
> ---
>
> ### **[Q2] Theorem 5.3**
> Theorem 5.3 proves **rationality agreement** $||u_{\mathrm{ind}} - u_{\mathrm{team}}|| -> 0$ (Sec. 5.2). When $u_{\mathrm{team}} = grad J$, this is the critical alignment mechanism that makes decentralized learning track the centralized objective and suppresses rotational “chasing” dynamics, explaining our stability/performance gains.
> For stationarity, Theorem 5.3 itself is an alignment statement; combined with the standard diminishing-step/noise conditions assumed in our convergence section, the resulting stable descent yields convergence to a first-order stationary set in the usual stochastic-approximation sense.
>
> ---
>
> ### **[Q3] Objective Relevance**
> $V$ is a direct measure of how far decentralized learning is from true team ascent (mismatch between local update directions and the team-gradient direction). Driving $V$ down is therefore not a cosmetic auxiliary objective: it is the mechanism that turns a game-like, rotational update field into a more potential-like, cooperative ascent dynamic. Empirically, whenever $V$ contracts, we consistently see higher return/success and stronger coordination signatures (higher alignment, lower GCR).
>
> ---
>
> ### **[Q4]–[Q5] Integration / comparison and large-scale benchmarks**
> **Integration / comparison:** HALyPO is a drop-in **update kernel**: it plugs into MAPPO/HAPPO by projecting the actor update, and into QMIX-style training via a differentiable team-return surrogate. For fair comparison we keep the backbone fixed and only swap the update kernel.
>
> **Heterogeneity:** HALyPO targets heterogeneous agents (no parameter sharing / mismatched policy classes) where decentralized gradients are structurally misaligned with the team's, HALyPO explicitly contracts this mismatch, a failure mode not directly addressed by standard MAPPO/QMIX.
>
> **SMAC and SMACv2:** We report mean win rates (%) below. HALyPO handles this large-scale benchmark and is competitive (4/5 wins, 1/5 on-par):
>
> | SMAC map | MAPPO | QMIX | HAPPO | HATRPO | HALyPO |
> | :--- | :---: | :---: | :---: | :---: | :---: |
> | `8m_vs_9m` | 86.3 | 89.1 | 82.7 | 90.8 | 91.7 |
> | `5m_vs_6m` | 74.9 | 76.2 | 76.5 | 74.0 | 76.8 |
> | `3s5z` | 94.8 | 88.7 | 95.3 | 92.6 | 96.1 |
> | `27m_vs_30m` | 80.2 | 49.3 | 75.6 | 90.4 | 92.3 |
> | `corridor` | 96.6 | 85.8 | 93.5 | 91.7 | 95.9 |
>
> ---
>
> ### **[Q6] Computational Complexity**
> HALyPO computes $h$ via double backprop (an HVP, no explicit Hessian), so the pure gradient step is about 2x heavier; in practice the end-to-end cost is much smaller in parallel simulators. In Isaac Lab, we measure 28.25 ms per step for HAPPO vs 35.72 ms for HALyPO (+26.4%), with <21.1% extra peak memory (see our response to Reviewer WVU6 [Q1]). Because HALyPO converges in fewer steps it reduces total wall-clock time to reach the plateau by 13.1% vs HAPPO. Scaling with agents is approximately linear in $D$ and thus typically linear in agent count for fixed per-agent network sizes (WVU6 [Q1]).
>
> ---
>
> ### **[Q7] Wording and Terminology**
> We appreciate this suggestion and will carefully standardize terminology in the final version, such as: Independent (decentralized) policy-gradient field $u_{\mathrm{ind}}$ (replacing “variational independent rationality”);  Joint team gradient field $u_{\mathrm{team}}$ (replacing “centralized greedy preferences”); Parameter Jacobian $H_{\mathrm{ind}}$ (replacing “co-evolutionary trajectory”)
>
> ---
>
> ### **[Q8] Ablation Enhancement**
> We appreciate the reviewer’s constructive suggestion. In the final manuscript, we will further expand ablations to incorporate using $u_{\mathrm{team}}$ only. For the case with Only $u_{\mathrm{ind}}$, it is actually covered by the HAPPO baseline (pure decentralized actor updates under CTDE), which is prone to uncoordinated gradient conflicts (high V).
>
> ---
>
> ### **[Q9] Impact Statement**
> In the final manuscript, we will thoroughly revise the Impact Statement to provide a balanced discussion of potential risks, such as safety issues in human–robot coupling and workforce considerations, etc.
>
> ---
>
> We sincerely thank the reviewer for the constructive feedback! We will open-source the project upon acceptance and hope that our clarifications support a more positive assessment of this work. Thank you!

---

> > ### Author Rebuttal · Reviewer_naJy · 2026-04-02
> >
> > My concerns have been well addressed.

---

> > > ### Author Response · Authors · 2026-04-03
> > >
> > > We are glad that all concerns are fully resolved! Thanks again for your insightful review!

---

### Official Review · Reviewer_rTmG · 2026-03-13

**Soundness:** 4
**Presentation:** 4
**Significance:** 4
**Originality:** 4
**Overall Recommendation:** 4
**Confidence:** 3

**Summary:**

This paper addresses the "rationality gap" (RG) in heterogeneous multi-agent reinforcement learning (MARL) applied to human-robot collaboration (HRC). The authors propose Heterogeneous-Agent Lyapunov Policy Optimization (HALyPO), which uses a Lyapunov potential defined on the parameter-space disagreement to stabilize decentralized policy learning. By framing the update as an optimal quadratic projection, the method aims to enforce monotonic contraction of the RG. The theoretical claims are supported by simulations in Isaac Lab and real-world experiments using a Unitree G1 robot.

**Compliance With Llm Reviewing Policy:**

Affirmed.

**Key Questions For Authors:**

1. The MARL policy runs at 2 Hz while the whole-body controller runs at 50 Hz. Could this introduce noticeable latency during rapid human motion changes?
2. Were multiple human partners used in the experiments? In HRC setups, performance gains can sometimes come from humans adapting to the robot policy.

**Limitations:**

The approach relies on assumptions (e.g., smoothness conditions and locally stationary partner behavior) that may be hard to guarantee in deep RL with humans in the loop. The evaluation also focuses on a specific task with a MoCap setup, so the generalization to other HRC scenarios or sensing conditions remains unclear.

**Strengths And Weaknesses:**

Strengths
This work presents a theoretically rigorous Lyapunov stability approach transposed from traditional state constraints to policy-parameter space for MARL’s non-symmetric Jacobian problem, with a scalable HVP double backpropagation implementation avoiding intractable full Hessian computation, validated via real-world humanoid deployment demonstrating strong sim-to-real transferability.
Weaknesses
1. HALyPO relies on second-order gradients (double backprop), which likely increases memory usage and wall-clock training time compared to first-order MARL methods. The paper does not report these metrics, so the practical efficiency is unclear.
2. The monotonic improvement guarantee depends on a step-size bound involving a global Lipschitz constant, which is typically unknown and difficult to estimate for deep networks.
3. The method assumes the partner policy is locally stationary when computing the rationality field. In real human-robot interaction, human behavior is often adaptive and non-stationary, which may affect gradient estimation.

---

> ### Author Rebuttal · Authors · 2026-03-30
>
> We sincerely thank the reviewer for the insightful comments. Below we address the concerns:
>
> ---
>
> ### **[Q1] Computational Efficiency**
> We thank the reviewer for these insightful comments. Deployability in parallel training environments was one of our primary objectives. HALyPO utilizes double back-propagation to compute the HVP, avoiding explicit Hessian computation and remaining practical. Based on our validation in Isaac Lab:
>
> * While HVP increases the gradient computation step ($T_{grad}$) by approximately 2x, this is diluted by environment sampling and inference costs in high-throughput settings. The total time per decision cycle ($T_{step}$) increased by only 26.4% (28.25 ms for HAPPO vs. 35.72 ms for HALyPO). Besides, we observe negligible additional memory overhead (<21.1%) due to HVP implementation via autograd without explicit Hessian materialization.
> * HALyPO optimizes the convergence path, reducing the GCR from 72.5% to 4.2%. It reaches the performance plateau in significantly fewer steps (Table 1). Consequently, HALyPO even achieves a 13.1% reduction in total wall-clock training time compared to HAPPO.
>
> Also, we have open-sourced the code architecture under anonymous conditions (provided in the response to reviewer WVU6) and will release the full code immediately upon acceptance for efficiency verification. Besides, we will include explicit memory overhead measurements in the final version.
>
> ---
>
> ### **[Q2] Step-Size Constraints**
>
> We wish to clarify that while our theoretical analysis establishes $L$ as a **sufficient condition** for stability, HALyPO does not require explicit prior knowledge of the global $L$ in practice. This is addressed through the following synergy:
>
> * By employing a **Trust Region (KL divergence)** constraint, we ensure policy updates remain within a local neighborhood where the local curvature is bounded, effectively bypassing the need for a global $L$ estimate.
> * HALyPO’s **quadratic projection** onto the stability half-space provides a wider convergence domain than standard first-order methods, significantly reducing sensitivity to exact constant estimation.
> * While $V(\theta)$ may exhibit minor initial fluctuations during exploration, the synergy of the trust region and **cosine annealing** ensures the update eventually aligns with the local curvature, driving the system toward monotonic dissipation.
>
> We will explicitly clarify in the final version that our stability certificate is maintained via these practical mechanisms without requiring a pre-defined global Lipschitz constant. We are grateful for the reviewer’s professional reminder!
>
> ---
>
> ### **[Q3] Non-Stationarity**
> We totally agree that human behavior is inherently adaptive. HALyPO is indeed designed to handle such non-stationarity:
>
> * Traditional decoupled methods (e.g., MAPPO, HAPPO, HATRPO) assume other agents remain fixed during local updates, leading to gradient bias and instability when partners exhibit non-stationarity. However, **HALyPO does not assume stationarity** but actively identifies and neutralizes its negative impacts. We enforce geometric constraints via optimal quadratic projection to transform the **oscillatory system into a dissipative one**.
> * In our experiments, HALyPO outperformed baselines particularly in **non-stationary scenarios**, as shown in the video results (in the reply to reviewer WVU6) and Figure 7~8, showing reduced oscillation and faster stabilization under changing partner behaviors.
>
> ---
>
> ### **[Q4] Control Frequency**
> Our hierarchical design decouples tactical coordination from high-frequency stabilization: Human intent changes typically occur at low frequency, and the 2 Hz update is **sufficient (based on experiments)** to capture and respond to these collaborative intent shifts of humans. Simulations and human participants reported no latency, as the WBC manages immediate physical coupling while the MARL manages the tactical trajectory.
>
> ---
>
> ### **[Q5] Participant Diversity**
> To exclude bias from human adaptation, we used a multi-user rotation protocol with three researchers who were instructed not to cater to the robot:
>
> * **Lead Experimentalist**: 3 tests for baseline consistency.
> * **Assistant Experimentalists**: 2 members conducted 1 test each to account for variability in gait and intent.
> * **Zero-Adaptation**: Participants were blind to algorithmic logic.
>
> ---
>
> ### **[Q6] Limitations**
> We thank the reviewer for raising these important points. While HALyPO is not tied to specific human scripts, we agree that broader generalization across sensing modalities remains to be validated. Future work will introduce **onboard vision and depth sensors** to verify generalization in non-MoCap environments.
>
> ---
>
> We sincerely commit to open-sourcing the entire project immediately upon acceptance. Should our responses clarify your concerns, we hope this helps support a more positive assessment of the work. Thank you for your time and consideration!

---

> > ### Author Rebuttal · Reviewer_rTmG · 2026-04-03
> >
> > My concerns have been addressed.

---

> > > ### Author Response · Authors · 2026-04-03
> > >
> > > We sincerely thank reviewer for acknowledging and happy to know that the concerns are resolved!

---

### Decision · Program_Chairs · 2026-04-30

**Decision:**

Accept (spotlight)

**Comment:**

The paper considers the problem of multi-agent reinforcement learning for human-robot collaboration. They analyze the problem of instability due to decentralized updates to cooperative policies, introduce a novel method to stabilize RL based on lyapunov analysis in policy space, and demonstrate this approach in a real robot experiment. Reviewers appreciated the importance of the problem considered, the theoretical contributions to a problem that is often approached heuristically, and the empirical results. Reviewers raised concerns about the presentation and the breadth of the experimental results. These were adequately addressed in the rebuttal and reviewers updated their scores accordingly.